# *OsXTH19* Overexpression Improves Aluminum Tolerance via Xyloglucan Reduction in Rice Root Cell Wall

**DOI:** 10.3390/plants14131912

**Published:** 2025-06-22

**Authors:** Akane Tatsumi, Teruki Nagayama, Ayumi Teramoto, Atsuko Nakamura, Ryusuke Yokoyama, Jun Furukawa, Hiroaki Iwai

**Affiliations:** 1Institute of Life and Environmental Sciences, University of Tsukuba, Tsukuba 305-8572, Ibaraki, Japan; 2National Agriculture and Food Research Organization, Tsukuba 305-8602, Ibaraki, Japan; 3Graduate School of Life Sciences, Tohoku University, Sendai 980-8578, Miyagi, Japan; ryusuke.yokoyama.d6@tohoku.ac.jp; 4Department of Biology, School of Biological Sciences, Tokai University, Sapporo 005-8601, Hokkaido, Japan

**Keywords:** Aluminum, xyloglucan, *OsXTH19*, *Oryza sativa*

## Abstract

Aluminum (Al) dissolves from soil at low pH and is absorbed by plants, inhibiting their growth. Since most of the Al absorbed by plants is present in the cell wall, it is thought that the binding of Al to cell wall polysaccharides alters the properties of the cell wall and inhibits cell elongation. However, it remains unclear in which component of the cell wall Al accumulates. In this study, we determined the distribution of Al in rice root cell wall fractions under Al stress conditions. The results show that Al accumulates predominantly in the hemicellulose fraction, with *star1* mutants accumulating significantly more Al than WT plants. An analysis of cell wall sugars revealed an increase in xyloglucan content under Al stress, which influenced the inhibition of root elongation. *OsXTH19*, a member of the xyloglucan endotransglucosylase/hydrolase (XTH) family, exhibits only xyloglucan endohydrolase (XEH) activity and lacks endotransglucosylase (XET) activity. *OsXTH19* overexpressor rice (*OsXTH19-OX*) enhances the degradation of xyloglucan. Furthermore, *OsXTH19-OX* rice with reduced xyloglucan levels exhibited reduced Al accumulation and enhanced root growth under Al stress.

## 1. Introduction

Aluminum (Al) toxicity is one cause of damage to plant cells and the inhibition of plant growth. Under acidic conditions, Al dissolves into its ionic form, which is toxic to plants [1]. Al toxicity has become a major cause of crop failure worldwide [2]. Soil pH affects aluminum compounds [3]. Acid soils (pH < 5) leach Al as the soluble Al^3+^ ion, and its uptake is one of the most potent inhibitors of plant growth [4]. Growth inhibition by Al has been reported in several plants, including rice, wheat, maize, tomato, and *Arabidopsis*. The mechanism of Al toxicity is complex, and how Al inhibits root elongation is poorly understood [5,6]. However, most Al-related events result from the binding of Al to extracellular and intracellular materials due to its high affinity for oxygen donor compounds. Al^3+^ has been reported to interact with negatively charged root surfaces and inhibit nutrient uptake in acidic soils [7]. Most root elongation-inhibiting Al is localized in the epidermis and external cell walls [8].

Previous studies have reported that the cell wall is the major site of Al accumulation [9]. The cell wall is the first site of contact with Al and plays an important role not only in the control of plant development but also in the recognition and tolerance of Al toxicity. For example, Clarkson reported that 85% to 90% of the total Al accumulated in barley roots is tightly bound to cell walls [10]. The main components of plant cell walls are three polysaccharides: cellulose, hemicellulose, and pectin. These polysaccharides determine cell shape and mechanical strength. The cell wall matrix, which includes pectin and hemicellulose polysaccharides, generally occupies 10–30% of the dry weight of plants. Pectin acts as a cement, binding plant cells together, while hemicellulose forms cross-links between cellulose fibers. Pectin was thought to be the major polysaccharide capable of binding Al because it is negatively charged and contains carboxyl groups that can directly bind Al [11]. However, in recent years, several studies have reported that it is hemicellulose that contributes significantly to the Al-binding capacity of cell walls [12]. This study suggests that the O-acetylation of xyloglucan in hemicellulose affects its Al-binding capacity, thereby influencing Al sensitivity [12]. Therefore, the results imply that one key mechanism by which plants cope with Al stress is by decreasing the Al content of cell walls through altering their components and properties. However, in rice, it is not clear where and how much Al accumulates in the cell wall and how this leads to growth inhibition. The mechanisms by which root-absorbed Al binds to cell wall components and what changes in cell wall components cause the inhibition of root elongation are still unclear. In this study, we used rice (*Oryza sativa*), which has relatively high Al tolerance [13,14]. We analyzed the Al responsibility of a WT with Al tolerance, the *star1* mutant as Al-sensitive rice, and xyloglucan endotransglucosylase/hydrolase (XTH) *OsXTH19-OX* as rice with suppressed xyloglucan accumulation. The *STAR1* gene encodes an ABC transporter, and its expression is controlled by the *ART1* transcription factor [15], which increases expression in response to Al stress. Since the *star1* mutant is strongly inhibited by Al, these genes are thought to contribute to Al tolerance in rice [16]. OsXTH19 has only xyloglucan endohydrolase (XEH) activity and no endotransglucosylase (XET) activity [17]. Therefore, in *OsXTH19-OX*, xyloglucan degradation proceeds, suppressing its accumulation and content. The XTH family is divided into three groups (I, II, and III) based on the structural features of each individual [14]. Group III comprises two distinct clades: Group III-A and Group III-B. According to crystallography data, Group III-A members are predicted to exhibit hydrolase activity rather than XET activity. *OsXTH19*, a Group III-A member, is specifically expressed in the elongating zones of roots [17]. Thus, this study focuses on the effects of aluminum on root elongation and its relationship with *OsXTH19*. In this study, we examined changes in Al accumulation sites and cell wall components in rice to elucidate the mechanism by which Al inhibits growth.

## 2. Results

### 2.1. The Quantification of the Amount of Al and Ca Present in Each Fraction of the Cell Wall

To determine the possible involvement of different cell wall components in the inhibition of root elongation by Al, we fractionated the major cell wall components into pectin, hemicellulose, and cellulose fractions. Al in the roots of each rice plant was quantified by inductively coupled plasma atomic emission spectroscopy (ICP-AES). When the amount of Al in 1 mg of dry root cell wall weight was quantified, *star1* contained 55% more Al than the WT under Al stress conditions. Furthermore, in both the WT and *star1*, about half of the Al present in the cell wall was in the hemicellulose fraction (Figure 1). In addition, to determine the background value of Al measurements that plants have prior to Al treatment, we quantified the Al contained in the seeds used in the experiment and found that approximately 1 mg of Al was detected per seed (Appendix A).

Similarly to Al quantification, ICP-AES quantified Ca in the roots of each rice plant. When the amount of Ca contained in 1 mg of root dry cell wall weight was quantified, similar amounts of Ca were detected in the WT and *star1* in the control conditions. In the hemicellulose fraction, there was a tendency for more Ca to be detected in *star1* than in the WT under Al stress conditions. About 45% more Ca was detected in the hemicellulose fraction than in the pectin fraction (Figure 2).

### 2.2. Analysis of Cell Wall Sugars in WT and Star1

Gas chromatography (GC) was used to measure cell wall constituent sugars in the trifluoroacetic acid (TFA)-soluble fraction, containing a large amount of hemicellulose. Fractionation with 2 M TFA can separate and extract a crystalline fraction composed mainly of cellulose fibers and an amorphous fraction consisting primarily of pectin and hemicellulose. The TFA-soluble fraction used this time is the fraction obtained after the fractional extraction of the EPG fraction that solubilizes pectin, so it is a fraction containing mainly hemicellulose cell wall components. As a result, in the hemicellulose fraction, the levels of xylose, glucose, and galactose, which make up xyloglucan, increased in the WT and *star1* due to Al stress. In addition, xylose and glucose were 20–30% more abundant in the WT than in *star1* with and without Al. On the other hand, the levels of glucuronic acid decreased in the WT and *star1* due to Al stress (Figure 3).

### 2.3. Measurement of Al Tolerance and Changes in Cell Wall Components in OsXTH19-OX

Previous reports examined the effects of the ectopic overexpression of *OsXTH19* [17]. Hara et al. generated ten independent *OsXTH19* overexpressor transgenic lines and selected two homozygous T2 lines, OX15 and OX18, that expressed relatively high levels of mRNA [17]. However, *OsXTH19* overexpression was more stable in the OX15 line than in the OX18 line. Thus, this experiment was performed using the OX15 line for *OsXTH19* overexpressor rice. In the experiments of this study, the levels of xylose and glucose in OsXTH19-OX rice were reduced by 19% and 43%, respectively, compared to the WT in the control condition (Figure 4). This reduction in glucose, which is the main chain of xyloglucan, was observed both in the presence and absence of Al. In addition, the level of galacturonic acid, the main component of pectin, was reduced by 52% compared to the WT under 100 µM Al conditions (Figure 4). This reduction in galacturonic acid was not observed in the absence of Al. Furthermore, *OsXTH19-OX* had less staining of pectin than the WT under control and Al stress conditions (Figure 5 and Appendix A). The amount of root elongation under Al stress conditions was measured, and the RRE value, which indicates the relative amount of root elongation, was calculated. As a result, when 100 µM Al stress was applied, root elongation was inhibited by 90% in the WT but only 72% in *OsXTH19-OX* (Figure 6). From the above, these results suggest that the reduction in root elongation inhibition in OsXTH19-OX increases with increasing Al stress concentration. Using ICP-AES, we compared the amount of Al present and accumulated in the cell wall region of the WT and *OsXTH19-OX*. As a result, about half of the Al present in the cell wall was accumulated in the TFA-soluble fraction, similar to the result in Figure 1. Under Al stress conditions, *OsXTH19-OX* contained 50% less Al than the WT (Figure 7).

## 3. Discussion

### 3.1. Reevaluating the Primary Target of Aluminum Toxicity and the Role of Hemicellulose in Rice Cell Walls

Whether the main target of Al toxicity is the cell wall or intracellular regions is still a matter of debate [5,18]. Since Al has a rather strong binding affinity for enzymatic reactions [5], Al likely targets multiple sites simultaneously. On the other hand, given that the cell wall is the main site of Al accumulation, it seems reasonable that the cell wall is the main target site of Al toxicity, especially when the treatment period is short. Indeed, there is ample experimental evidence that Al-induced root growth inhibition is associated with the disruption of cell wall function [18]. For example, Al treatment has been shown to affect the growth of many plant species such as pumpkin [9], wheat [19], and rice [12,13,14]. Li et al. reported that the mechanism of Al-induced root growth inhibition in maize is the binding of Al to pectin [7]. However, the biochemical analysis results of the cell wall underlying the inhibition of root elongation in response to Al treatment are not clear. Therefore, pectin has long been recognized as the major component of Al binding that inhibits root elongation, but this is not shown in the study of Nagayama et al. Our results indicate that hemicellulose is the major contributor to Al accumulation in the rice cell wall. Al toxicity is characterized by its short-term effects. One example of these effects is the inhibition of root elongation. The results of this study were obtained within 48 h of aluminum treatment. If the treatment had been carried out for a longer period, a feedback mechanism may have occurred to alleviate the effects. However, this report is limited to findings within 48 h.

### 3.2. Accumulation of Aluminum and Calcium in Cell Wall and Their Roles in Aluminum Tolerance

Low-Al-tolerance strains and the *star1* mutant have been reported to accumulate more Al [15,20], and we obtained similar results (Figure 1 and Figure 7). The background level of Al measurement derived from seeds that plants have before Al treatment is considered to be in the range of 0 to 1 µg per 1 mg of dry cell wall. About half of the Al present in the cell wall was contained in the TFA-soluble fraction, which is rich in hemicellulose (Figure 1). Several previous studies have also reported that hemicellulose adsorbs Al [14,21], suggesting that the Al absorbed by plants is mainly accumulated in the hemicellulose region of the cell wall. Some of the hydrogen atoms contained in the polysaccharides of hemicellulose form hydrogen bonds with cellulose, thereby cross-linking the cellulose in hemicellulose. Al^3+^ can accumulate by forming hydrogen bonds with hydroxyl groups in cellulose and hemicellulose regions without side chains. Even in *Arabidopsis* [12] and pumpkin [9], which have lower hemicellulose content than rice, hemicellulose has been confirmed as the major Al accumulation site, and its role in Al tolerance can be considered important. The EPG fraction, which contains a large amount of pectin, accounted for 30 to 40% of the Al accumulated in the cell wall (Figure 1). The main chain of pectin consists mainly of galacturonic acid, a type of uronic acid. It is then demethylated by pectin methylesterase (PME), exposing the carboxyl group of galacturonic acid, which is thought to become negatively charged, making it easier to bind to cations such as Ca^2+^ [20]. Normally, carboxyl groups gel by forming a cross-linked structure with Ca^2+^, which imparts viscoelasticity to the cell wall [22]. However, pectin has a higher affinity for Al^3+^ than for Ca^2+^, and Al^3+^ also has a high adsorption capacity for pectin [21]. Ca is an essential macronutrient for plants and plays an important role in plant development and stress responses as a structural component in cell walls and membranes and as an intracellular messenger in the cytoplasm [23]. It has been shown that Ca functions to reduce oxidative stress and alleviate heavy metal toxicity [4,10,24] and that the elongation of rice roots and aerial parts is significantly suppressed under non-Ca-supplemented conditions [25]. Previous studies have observed that Al treatment inhibits Ca uptake and significantly reduces the amount of Ca transported to aerial parts [25]. These results suggest that under Al stress conditions, Ca transport to the aerial parts is inhibited, so both Al and Ca are present in large amounts. In *star1*, where more Ca was detected in the roots than in the WT, it is possible that Ca transport was more inhibited than in the WT (Figure 2).

### 3.3. Xyloglucan Accumulation and Its Role in Aluminum Binding and Cell Wall Modification in Rice

Twenty to thirty percent of the Al accumulated in the cell walls was contained in the TFA-insoluble fraction, which contains a large amount of cellulose (Figure 1). Cellulose is a linear polysaccharide highly condensed with β-D-glucose. This main chain forms microfibrils through intramolecular and intermolecular hydrogen bonds, giving it a strong structure. As in the case of hemicellulose, Al^3+^ is thought to accumulate by forming hydrogen bonds with cellulose. However, because the bonds between celluloses are strong, there are few binding sites for Al^3+^ to bind to cellulose, which may have reduced the amount of Al accumulated in cellulose. Hemicellulose in grasses is mainly composed of xylan, MLG, and xyloglucan, each with distinct sugar compositions. [26,27]. An analysis of the constituent sugars of the TFA-soluble fraction containing a large amount of hemicellulose showed that xylose, glucose, and glucuronic acid were the major components (Figure 3 and Figure 4). It also contained a large amount of galacturonic acid, which is the main component of pectin, and it is suggested that the TFA-soluble fraction contained pectin with a good ability to bind to the cell wall (Figure 3). Among the constituent sugars of hemicellulose in pumpkin roots, an increase in the amount of glucose and xylose was observed due to Al treatment, which is thought to contribute to the increase in xyloglucan content [9,28]. These results suggest that Al stress increases xyloglucan levels in hemicellulose. XTH is an important enzyme involved in xyloglucan metabolism (Figure 4). The XTH enzyme is involved in cell wall elongation through endohydrolase (XEH) activity, which hydrolyzes xyloglucan, and endotransglucosylase (XET) activity, which rearranges xyloglucan [17,29]. In *Arabidopsis* mutants in which XTH31, which contributes significantly to xyloglucan transfer activity, is deleted and xyloglucan accumulation is suppressed, the amount of Al accumulated in roots is reduced, and high Al tolerance is obtained [30]. It has also been reported that when XET activity is inhibited by Al stress, the cleavage of the xyloglucan backbone and the subsequent binding of new xyloglucan chains are prevented, and cell wall elongation is inhibited [12]. While xyloglucan is the major component in the cell wall of dicots, the amount of xyloglucan in the rice cell wall is very low. It has been discussed whether xyloglucan is, in actuality, the substrate for all XTH in monocots [17,31,32,33]. Despite the difference in the amount of xyloglucan between monocots and dicots, rice is known to have the same number of XTH genes as *Arabidopsis* [34], and xyloglucans may also play an important role in regulating cell wall properties in rice. Xyloglucan is thought to act as a tether between cellulose microfibrils in the primary cell wall, limiting cell wall loosening [33,34]. However, the degradation of xyloglucan alone by specific endoglucanases does not loosen the wall; only enzymes targeting both xyloglucan and cellulose are effective [32,33]. These results suggest that xyloglucans are intertwined with cellulose microfibrils, forming regions that require enzymes with both cellulase and xyloglucanase activities to degrade. In addition, small amounts of xyloglucan adhere to adjacent cellulose microfibrils, suggesting that cell wall extensibility may be controlled in this limited region (biomechanical hotspot) [35]. Furthermore, observations of NMR spectra have confirmed that a complex is formed between xyloglucan and Al and that Al can bind to xyloglucan [36]. The OsXTH19 used in this study has only hydrolytic activity. Therefore, in *OsXTH19-OX*, the degradation of xyloglucan proceeds, suppressing its accumulation and content (Figure 4). In this *OsXTH19-OX*, the elongation inhibition conducted by Al was suppressed (Figure 6), and the amount of Al accumulated tended to be low, suggesting that xyloglucan is the target site of Al (Figure 7).

### 3.4. Reduced Xyloglucan Content Enhances Aluminum Tolerance Despite Lower Pectin Levels

Nagayama et al. reported that rice varieties with high Al tolerance increase the amount of water-soluble pectin secreted from their roots when they receive Al. This pectin acts as a barrier and reduces toxicity by preventing Al from adsorbing to hemicellulose, which is the active site of Al and one of the cell wall components. In contrast, in *star1*, which has low Al tolerance, pectin was not secreted from the roots even when exposed to Al stress, and Al accumulated in the cell wall [14,37,38]. However, in *OxXTH19-OX*, which has a higher tolerance for Al than the WT, the amount of pectin was less than that of the WT when treated with Al (Figure 4 and Figure 5). It was suggested that even if there was not a lot of pectin, Al tolerance was high when the amount of xyloglucan, the target of Al, was low. Since the build-up of Al in the roots was blocked in *OsXTH19-OX*, which has a low amount of xyloglucan, it is possible that the direct build-up of Al in xyloglucan affects cell wall growth and stops root growth. Factors important for cell growth that work by binding to xyloglucan, such as XTH and expansin, may not work as well when Al is present in xyloglucan. It has also been reported that monocotyledonous plants can tolerate Al [13,38]. This suggests that one reason for this is that the cell walls of monocotyledonous plants have a lower amount of xyloglucan compared to the cell walls of dicotyledonous plants.

## 4. Materials and Methods

### 4.1. Plant Material and Growth Conditions

The wild-type (Oryza sativa cv. Koshihikari) and the star1 mutant (Koshihikari background) [15] were used in the experiments. The wild-type (Oryza sativa cv. Nipponbare) and the *OsXTH19-OX* (Nipponbare background, OX15 line) [17] rice strains were also used in the experiments. Seedlings were submerged in ion exchange water at 30 °C for 3 days, followed by 3 days in 1.0 mM CaCl_2_, pH 4.5. Grown seedlings were treated with or without Al for 1 day. Free Al activity was determined by GEOCHEM-EZ software (version 1.0) [39], ranging between 76.57% and 78.28%. Plants were grown at 30 °C under continuous light of 250 μmol m^−2^ s^−1^. To calculate root elongation during Al treatment, root length was measured before and after Al treatment using a ruler. The relative root elongation, RRE (%) = (root growth in each Al condition)/(root growth in the control) × 100, was calculated to compare root elongation and Al tolerance between the different plants. All experiments were performed at least five times with independent biological replicates.

### 4.2. Cell Wall Analysis

Root tips (0–1 mm) from three seedlings were cut with a razor and collected in a 2.0 mL tube as a cell wall sample. Samples were frozen in liquid nitrogen and crushed with a pestle. Cell wall extraction and analysis were performed according to Sumiyoshi et al. with slight modifications [40]. Dry cell wall material (2 mg) was treated with endo-polygalacturonase (EPG, Megazyme, Ireland), a pectin hydrolase, and the mixture was stirred for 16 h at 30 °C and 40 rpm in a stirrer as described in T. Ishii et al. [41]. The sample was then centrifuged, and the supernatant was used as the EPG fraction; the pellet was hydrolyzed in 2 M trifluoroacetic acid (TFA; Wako, Osaka, Japan) at 121 °C for 2 h, and the pellets were also hydrolyzed in 72% H_2_SO_4_ at room temperature for 2 h. Each fraction was treated with methanol–hydrogen chloride, and the resulting methyl glycosides were converted to trimethylsilyl derivatives and analyzed by gas–liquid chromatography (GC-2010; Shimadzu, Kyoto, Japan). The hexose sugar content and uronic acid content were determined by the anthrone method and the meta-hydroxybiphenyl method, respectively.

### 4.3. Quantification of Al and Ca

The fractionated EPG fraction and TFA-soluble fraction were poured into a screw-capped test tube, and 5 mL of high-purity 60% HNO_3_ for ICP was added. The test tube was placed on a heater (TAITEC Dry Thermo Unit DTU-2C), then covered with a marble and left at room temperature for 18 h. The glass test tube, measuring cylinder, and marble used in this experiment were all washed with 1% HNO_3_ overnight. Subsequently, the heater was set to 90 °C, and the samples were heated for 1 h, followed by heating at 120 °C for 3 h. After cooling to room temperature, the contents were transferred to 15 mL tubes. Subsequently, 10 mL of Milli-Q water was added to 1 mL of this sample to dilute it to 6% HNO_3_. The sample was filtered using a 10 mL Termo syringe (TERMO SS-10SZ) and a 0.22 µm syringe filter (Rephile 1-1517-02). The quantification of Al and Ca in the nitric acid-digested sample was performed using an ICPS-8100 (Shimadzu, Kyoto, Japan). Calibration curve solutions (0, 0.1, 1.0, 5.0 ppm) were prepared using Al and Ca standard solutions. Measurements were conducted at wavelengths with minimal co-existing elements and maximum intensities of Al 167.079 nm and Ca 393.366 nm with the ICPS-8100.

### 4.4. Staining of Total Pectin with Ruthenium Red

Methylesterified pectin in roots was saponified with 0.1 N NaOH in a 50 mL centrifuge tube for 1 min to remove methyl groups from pectin and to convert methylesterified pectin to demethylesterified pectin. Following saponification, roots were washed with ion exchange water; then all pectins were stained with ruthenium red [13]. Sample roots were stained with 0.01% (*w*/*w*) ruthenium red in a 50 mL centrifuge tube for 5 min to detect the dimethylesterified pectin in the roots. The stained root was rinsed with ion exchange water [14].

### 4.5. Statistical Analysis

The data were expressed as the mean values ± SD taken from 4 to 9 independent biological experiments. The experimental data of the samples were statistically analyzed through a one-way analysis of variance (ANOVA) with Tukey’s post hoc test using Statistica 13.1 software (StatSoft, Inc., Tulsa, OK, USA). The results with a *p*-value ≤ 0.05 and a *p*-value ≤ 0.01 were considered statistically significant.

## Figures and Tables

**Figure 1 plants-14-01912-f001:**
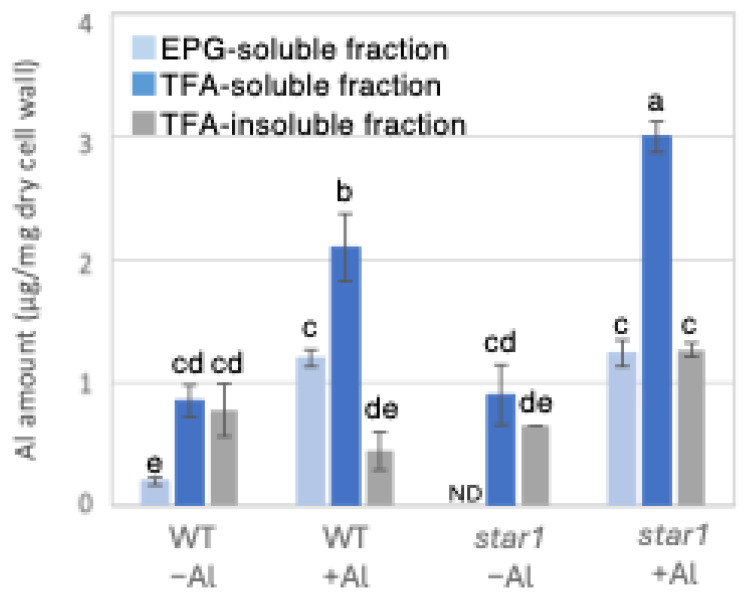
Al content in different cell wall fractions of WT and aluminum-sensitive mutant *star1*. Plants were treated without or with Al (0: −Al or 100 µM AlCl_3_: +Al), and cell wall fractions were extracted from roots and fractionated into different residues (endo-polygalacturonase (EPG)-soluble fraction as pectic fraction, trifluoroacetic acid (TFA)-soluble fraction as hemicellulosic fraction, and TFA-insoluble fraction as cellulosic fraction). After extraction of Al with HNO_3_, Al content was determined by ICP-AES. Data represent means of independent biological replicates ± standard deviation (n = 5). Different letters in each panel indicate significant differences at *p* < 0.05 (Tukey’s test). ND, not detected.

**Figure 2 plants-14-01912-f002:**
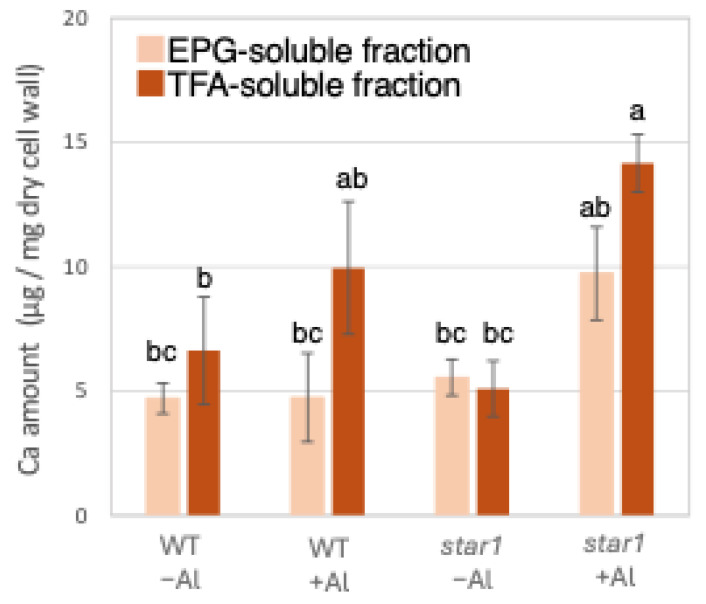
Ca content in different cell wall fractions of WT and aluminum-sensitive mutant *star1*. Plants were treated without or with Al (0: −Al or 100 µM AlCl_3_: +Al), and cell wall fractions were extracted from roots and fractionated into different residues (EPG-soluble fraction as pectic fraction and TFA-soluble fraction as hemicellulosic fraction). After extraction of Al with HNO_3_, Al content was determined by ICP-AES. Data represent means of independent biological replicates ± standard deviation (n = 5). Different letters in each panel indicate significant differences at *p* < 0.05 (Tukey’s test).

**Figure 3 plants-14-01912-f003:**
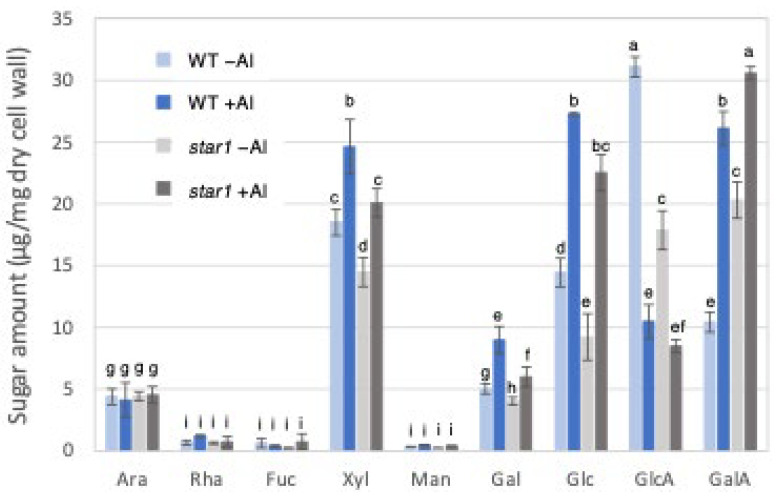
Monosaccharide composition of TFA-soluble fractions of WT and aluminum-sensitive mutant *star1*. Monosaccharide composition of TFA-soluble fractions in roots from WT (cv Koshihikari) and *star1*. Plants were treated without or with Al (0: −Al or 100 µM AlCl_3_: +Al). SD is given in parentheses. Different letters in each panel indicate significant differences at *p* < 0.05 (Tukey’s test). [Data represent means of independent biological replicates ± standard deviation. n = 6 (WT), and n = 6 (*star1*)].

**Figure 4 plants-14-01912-f004:**
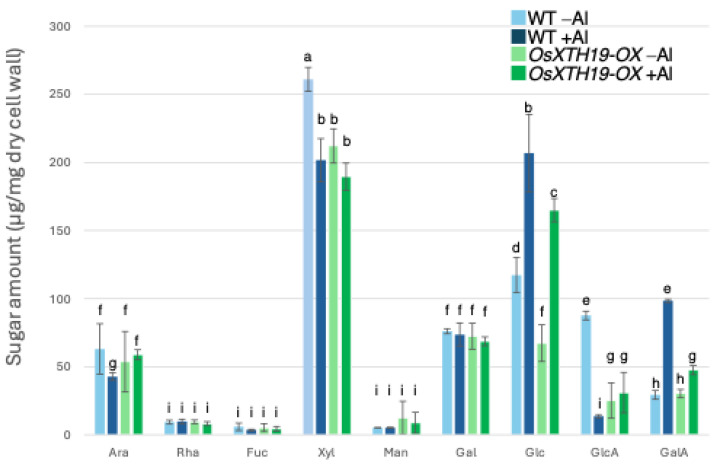
Monosaccharide composition of TFA-soluble fractions of WT and OsXTH19-OX. Monosaccharide composition of TFA-soluble fractions in roots from WT (cv Nipponbare) and *OsXTH19-OX*. SD is given in parentheses. Plants were treated without or with Al (0: −Al or 100 µM AlCl_3_: +Al). Different letters in each panel indicate significant differences at *p* < 0.05 (Tukey’s test). [Data represent means of independent biological replicates ± standard deviation. n = 6 (WT), and n = 7 (*OsXTH19-OX*)].

**Figure 5 plants-14-01912-f005:**
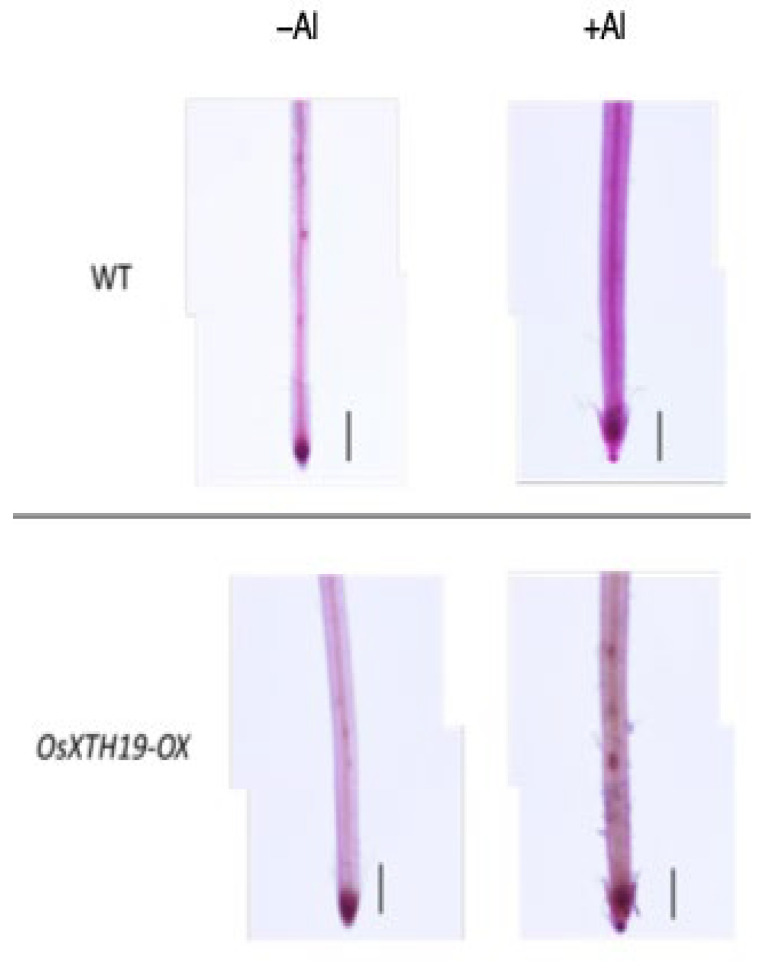
Pectin staining in the roots of the WT and *OsXTH19-OX* seedlings treated without or with Al (0 or 100 µM AlCl_3_). Roots were stained with 0.01% ruthenium red for 5 min after saponification (0.1 N NaOH, 1 min). The experiments were performed at least five times with similar results. Bars = 0.1 mm.

**Figure 6 plants-14-01912-f006:**
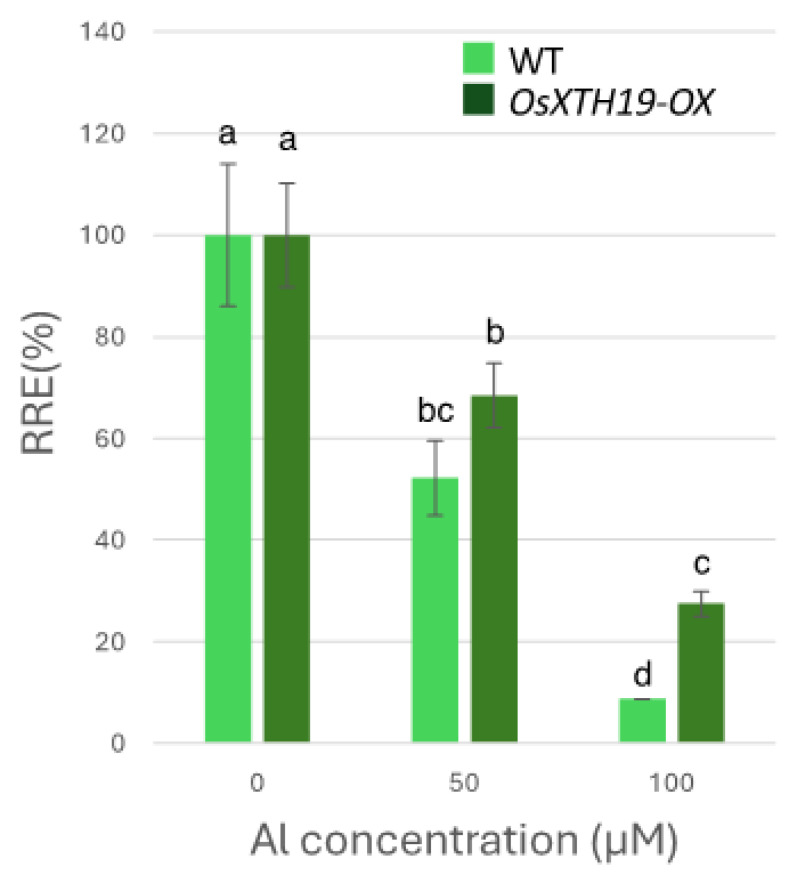
Effects of Al on root growth of WT and *OsXTH19-OX*. Plants were treated with Al (0, 50, 100 µM AlCl_3_). Seedling root length was measured before and after Al treatment, and amount of root elongation was determined. Root elongation differed significantly between WT and *OsXTH19-OX* rice under 100 µM Al treatment (*p* < 0.05, Student’s *t*-test). Data are means ± SDs, n = 12. Different letters in each panel indicate significant differences at *p* < 0.05 (Tukey’s test).

**Figure 7 plants-14-01912-f007:**
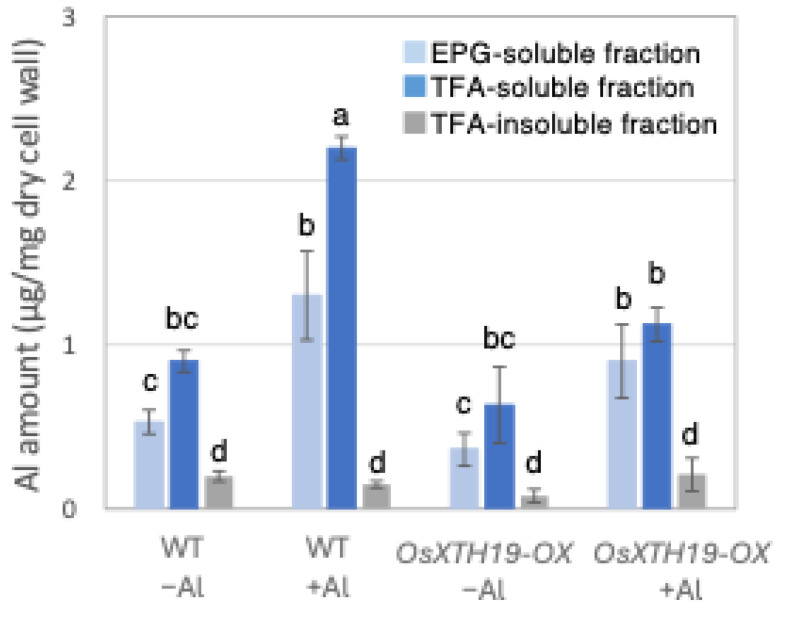
Al content in different cell wall fractions of WT and *OsXHT19-OX*. Plants were treated without or with Al (0: −Al or 100 µM AlCl_3_: +Al), and cell wall fractions were extracted from roots and fractionated into different residues (endo-polygalacturonase (EPG)-soluble fraction as pectic fraction, trifluoroacetic acid (TFA)-soluble fraction as hemicellulosic fraction, and TFA-insoluble fraction as cellulosic fraction). After extraction of Al with HNO_3_, Al content was determined by inductively coupled plasma atomic emission spectrometry. Data represent means of independent biological replicates ± standard deviation (n = 5). Different letters in each panel indicate significant differences at *p* < 0.05 (Tukey’s test).

## Data Availability

The original contributions presented in this study are included in the article/Appendix A. Further inquiries can be directed to the corresponding authors.

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
