# Peer review of "OsXTH19* Overexpression Improves Aluminum Tolerance via Xyloglucan Reduction in Rice Root Cell Wall"

_plants, 2025, doi:10.3390/plants14131912_

Round 1

Reviewer 1 Report

Comments and Suggestions for Authors

In this study the authors compare the distribution of aluminum (Al) in various polysaccharide components of cell walls derived from roots of rice genotypes that differed in Al tolerance. The main finding was that reduced xyloglucan levels in transgenic rice overexpressing xyloglucan endohydrolase activity was associated with an increase in Al tolerance.

Major points

  1. The authors need to provide a better description of the germplasm. First, they should describe the overexpression lines more adequately in the Introduction (lines 54 to 56) and state that the particular overexpression line used in their study was previously described by Hara etal. 2017. In particular this should be detailed in Materials and Methods specifying which particular line was used. It would have been preferable if they had used both homozygous lines described by Hara etal. 2017 instead of just a single line.
  2. Take care when describing the various Figures and making claims about differences when the values are not statistically significant. For example, the text claims that 67% more Ca was detected in star1 than WT (line 96) when the statistical analysis does not show a difference (both share the same letter “a”). Carefully check other Figures to avoid the same problem, for example lines 90-92 seems to make a similar error.
  3. Is a bar for the EPG fraction of star1 -Al missing from Figure 1?
  4. Figures 3 and 4 lack statistical analyses (ie, letters indicating significant differences).
  5. Is it possible to quantify the staining intensity of various sections of the roots showing in Figure 5? This would be far preferable than showing an image of a single root for each treatment. An image of the roots could still be shown but as support for the quantified data.
  6. Figure 6- a significant difference between genotypes is only apparent at 100 uM Al and not at 50 uM as stated. Here the use of two independent transgenic lines would provide added strength to the main conclusion since with the current data it is only at a single Al concentration where the genotypes differ.

Minor points.

  1. Take care when using acronyms. For example, ICP-AES is defined early in the text yet is spelled out in full in various Figures. Check other acronyms to avoid the same problem.
  2. Don’t assume that readers will necessarily understand that OsXTH18-OX means a transgenic line overexpressing the XTH gene. Describe the line and genes more fully as discussed under Major points.
  3. Take care to use italics for star1consistently. In some cases it is written in italics and often it is plain text.
  4. Lines 123-124: more correctly it is the activity of the enzyme that was at a high level and not the expression in cell walls.
  5. Materials and Methods (line 289)- should be 1.0 mM CaCl2?

Author Response

Dear Reviewer 1,

We are grateful to the editor and reviewers for their constructive and helpful comments, which greatly helped us improve our manuscript titled “OsXTH19 Overexpression Improves Aluminum Tolerance via Xyloglucan Reduction in Rice Root Cell Wall” (plants-3689902). As indicated in the following responses, we have considered all these comments in preparing this resubmitted manuscript. Detailed responses to each comment are listed below.

Reviewer 1

In this study the authors compare the distribution of aluminum (Al) in various polysaccharide components of cell walls derived from roots of rice genotypes that differed in Al tolerance. The main finding was that reduced xyloglucan levels in transgenic rice overexpressing xyloglucan endohydrolase activity was associated with an increase in Al tolerance.

 Thank you for your review. We have thoroughly revised our manuscript.

Major points

  1. The authors need to provide a better description of the germplasm. First, they should describe the overexpression lines more adequately in the Introduction (lines 54 to 56) and state that the particular overexpression line used in their study was previously described by Hara etal. 2017. In particular this should be detailed in Materials and Methods specifying which particular line was used. It would have been preferable if they had used both homozygous lines described by Hara etal. 2017 instead of just a single line.

Thank you for your comments. We fully agree with your advice. According to the reviewer’s advice, we have revised the Introduction, Results, and Methods sections as follows.

Introduction

(L75)

The XTH family is divided into three groups (I, II, and III) based on the structural features of each individual [14]. Group III comprises two distinct clades: Group III-A and Group III-B. According to crystallography data, Group III-A members are predicted to exhibit hydroase activity rather than XET activity. OsXTH19, a Group III-A member, is specif-ically expressed in the elongating zones of roots [17]. Thus, this study focuses on the effect of aluminum on root elongation and its relationship to OsXTH19. In this study, we examined changes in Al accumulation sites and cell wall components in rice to elucidate the mechanism by which Al inhibits growth.

Results

(L144)

2.3. Measurement of Al tolerance and changes in cell wall components in OsXTH19-OX

Previous reports examined the effects of the ectopic overexpression of OsXTH19 [17]. Hara et al. generated ten independent OsXTH19overexpressor transgenic lines and selected two homozygous T2 lines, OX15 and OX18, that expressed relatively high levels of mRNA [17]. However, OsXTH19 overexpression was more stable in the OX15 line than in the OX18 line. Thus, this experiment was performed using the OX15 line for OsXTH19 overexpressor rice.

Materials and Methods

(L304)

the OsXTH19-OX (Nipponbare background, OX15 line) [17] rice strains were also used in the experiments.

  1. Take care when describing the various Figures and making claims about differences when the values are not statistically significant. For example, the text claims that 67% more Ca was detected in star1 than WT (line 96) when the statistical analysis does not show a difference (both share the same letter “a”). Carefully check other Figures to avoid the same problem, for example lines 90-92 seems to make a similar error.

We have corrected the errors as follows.

Similar to Al quantification, ICP-AES quantified Ca in the roots of each rice plant. When the amount of Ca contained in 1 mg of root dry cell wall weight was quantified, similar Ca were detected in the WT and star1 in the control conditions. in the hemicellulose fraction. About 2.3 times more Ca was detected in star1 than in the control under Al stress conditions. About 45% more Ca was detected in the hemicellulose fraction than in the pectin fraction. Under control conditions, there was almost no difference in the amount of Ca between WT and star1, whereas under Al stress conditions, there was a tendency for more Ca to be detected in star1 than in WT (Figure 2).

  1. Is a bar for the EPG fraction of star1 -Al missing from Figure 1?

Thank you for your comment. Aluminum of this fraction was not detected, so we changed the sentences as follows.

Figure 1. Al content in different cell wall fractions of WT and the aluminum-sensitive mutant star1. Plants were grown in 1.0 mM CaCl2 at pH 4.5 and then treated without or with Al (0: -Al or 100 µM AlCl3: +Al), and cell wall fractions were extracted from roots and fractionated into different residues (endo-polygalacturonase (EPG)-soluble fraction as pectic fraction, trifluoroacetic acid (TFA)-soluble fraction as hemicellulosic fraction, and TFA-insoluble fraction as cellulosic fraction). After extraction of Al with HNO3, Al content was determined by inductively coupled plasma-atomic emission spectrometry. Data represent means of independent biological replicates ± standard deviation (n = 5). Different letters in each panel indicate significant differences at p < 0.05 (Tukey's test). ND, not detected.

  1. Figures 3 and 4 lack statistical analyses (ie, letters indicating significant differences).

According to the reviewer’s comments, we added the statistical analysis and revised Figures 3 and 4.

  1. Is it possible to quantify the staining intensity of various sections of the roots showing in Figure 5? This would be far preferable than showing an image of a single root for each treatment. An image of the roots could still be shown but as support for the quantified data.

According to the reviewer’s comments, we added the NEW supplementary Figure S2.

  1. Figure 6- a significant difference between genotypes is only apparent at 100 uM Al and not at 50 uM as stated. Here the use of two independent transgenic lines would provide added strength to the main conclusion since with the current data it is only at a single Al concentration where the genotypes differ.
  2.  

According to the reviewer’s comments, we added the sentences in the Results as follows.

(L144)

2.3. Measurement of Al tolerance and changes in cell wall components in OsXTH19-OX

Previous reports examined the effects of the ectopic overexpression of OsXTH19 [17]. Hara et al. generated ten independent OsXTH19overexpressor transgenic lines and selected two homozygous T2 lines, OX15 and OX18, that expressed relatively high levels of mRNA [17]. However, OsXTH19 overexpression was more stable in the OX15 line than in the OX18 line. Thus, this experiment was performed using the OX15 line for OsXTH19 overexpressor rice.

Minor points.

  1. Take care when using acronyms. For example, ICP-AES is defined early in the text yet is spelled out in full in various Figures. Check other acronyms to avoid the same problem.

Thank you for your advice. We reconfirmed that in the manuscript.

  1. Don’t assume that readers will necessarily understand that OsXTH18-OX means a transgenic line overexpressing the XTH gene. Describe the line and genes more fully as discussed under Major points.

According to the reviewer’s advice, we have revised the abstract as follows.

Abstract: Aluminum (Al) dissolves from soil at low pH and is absorbed by plants, inhibiting their growth. Since most of the Al absorbed by plants is present in the cell wall, it is thought that binding of Al to cell wall polysaccharides alters the properties of the cell wall and inhibits cell elongation. However, it remains unclear in which component of the cell wall Al accumulates. In this study, we determined the distribution of Al in rice root cell wall fractions under Al stress conditions. The results show that Al accumulates predominantly in the hemicellulose fraction, with star1 mutants accumulating significantly more Al than WT plants. Analysis of cell wall sugars revealed an increase in xyloglucan content under Al stress, which influenced the inhibition of root elongation. OsXTH19, a member of the xyloglucan endotransglucosylase/hydrolase (XTH) family, exhibits only xyloglucan endohydrolase (XEH) activity and lacks endotransglucosylase (XET) activity. OsXTH19overexpressor rice (OsXTH19-OX) enhances the degradation of xyloglucan. Furthermore, OsXTH19-OX rice with reduced xyloglucan levels exhibited reduced Al accumulation and enhanced root growth under Al stress.

  1. Take care to use italics for star1consistently. In some cases it is written in italics and often it is plain text.

Thank you for your advice. We have made the necessary changes to the manuscript and have changed the text to italics in red.

  1. Lines 123-124: more correctly it is the activity of the enzyme that was at a high level and not the expression in cell walls.

We have corrected the errors as follows.

(L142)

2.3. Measurement of Al tolerance and changes in cell wall components in OsXTH19-OX

Previous reports examined the effects of the ectopic overexpression of OsXTH19 [17].

  1. Materials and Methods (line 289)- should be 1.0 mM CaCl2?

We have corrected the errors as follows.

(L305)

Seedlings were submerged in ion exchange water at 30 °C for 3 days, followed by 3 days in 1.0 mM CaCl2,

We believe our manuscript now meets standards for publication. I am looking forward to hearing from you soon.

Sincerely yours,

Hiroaki Iwai

Tokai University

Department of Biology, School of Biological Sciences,

Sapporo, Hokkaido 005-8601, Japan

iwai.hiroaki.gb@tokai.ac.jp

Reviewer 2 Report

Comments and Suggestions for Authors

In the manuscript named “OsXTH19 Overexpression Improves Aluminum Tolerance via Xyloglucan Reduction in Rice Root Cell Wall”, authors have investigated star1 mutation lines and OsXTH19-OE line under Al stress condition, and they found OsXTH19 with enhancing the degradation of xyloglucan to reduce Al accumulation and improve Al tolerance. These findings would be helpful for rice production and crops’ protection in future. However, there were some comments about it.

  1. Authors didn’t clearly describe plant material in method sections, there were some fragmentary descriptions in introduction or results sections, but missing in method sections, please collect these descriptions in method section.
  2. Authors have introduced two genes, STAR1 and OsXTH19, but only described some traits or performances of mutant lines from star1, not clearly analyzed the results. In addition, there was no crosslinking about two genes in results or experiments.
  3. The “Monosaccharide composition” were widely differencing between two experiments, see figure 3 and figure 4, the WT lines would be same or almost same, but they were some big differences between two detections, please check them.
  4. In Pectin staining, the star1 mutant lines were missing, it would be important supplements for authors suppose.
  5. Authors thought OsXTH19-OE would reduce Al accumulation by transporting it, such as STAR1 gene, authors could detect its expression in WT and OsXTH19-OE lines to confirm its function pathway.
  6. “star1 mutant” in line 59 should be italic, etc.
  7. Some description in figures could be removed to method section, such as “Plants were grown in 1.0 mM CaCl2 at pH 4.5 and then treated without or with Al (0: -Al or 100 μM AlCl3: +Al),”, etc.

Author Response

Dear Reviewer 2,

We are grateful to the editor and reviewers for their constructive and helpful comments, which greatly helped us improve our manuscript titled “OsXTH19 Overexpression Improves Aluminum Tolerance via Xyloglucan Reduction in Rice Root Cell Wall” (plants-3689902). As indicated in the following responses, we have considered all these comments in preparing this resubmitted manuscript. Detailed responses to each comment are listed below.

Reviewer 2

In the manuscript named “OsXTH19 Overexpression Improves Aluminum Tolerance via Xyloglucan Reduction in Rice Root Cell Wall”, authors have investigated star1 mutation lines and OsXTH19-OE line under Al stress condition, and they found OsXTH19 with enhancing the degradation of xyloglucan to reduce Al accumulation and improve Al tolerance. These findings would be helpful for rice production and crops’ protection in future. However, there were some comments about it.

Thank you for your review. We have thoroughly revised our manuscript.

  1. Authors didn’t clearly describe plant material in method sections, there were some fragmentary descriptions in introduction or results sections, but missing in method sections, please collect these descriptions in method section.

Thank you for your comments. We fully agree with your advice. According to the reviewer’s advice, we have revised the Methods sections as follows.

  1. Materials and Methods

4.1. Plant Material and Growth Conditions

The wild-type (Oryza sativa cv. Koshihikari) and the star1 mutant (Koshihikari background) [15] were used in the experiments. The wild-type (Oryza sativa cv. Nipponbare) and the OsXTH19-OX (Nipponbare background, OX15 line) [17] rice strains were also used in the experiments. Seedlings were submerged in ion exchange water at 30 °C for 3 days, followed by 3 days in 1.0 mM CaCl2, pH 4.5. Grown seedlings were treated with or without Al for 1 day. Free Al activity was determined by GEOCHEM-EZ software [39], ranging between 76.57% to 78.28%. Plants were grown at 30 °C under continuous light of 250 μmol m-2 s-1. To calculate root elongation during Al treatment, root length was measured before and after Al treatment using a ruler. The relative root elongation, RRE (%) = (root growth in each Al condition)/(root growth in the control) × 100, was calculated to compare the root elongation and the Al tolerance between the different plants. All experiments were performed at least five times with independent biological replicates.

4.2. Cell Wall Analysis

Root tips (0-1 mm) from three seedlings were cut with a razor and collected in a 2.0 ml tube as a cell wall sample. Samples were frozen in liquid nitrogen and crushed with a pestle. Cell wall extraction and analysis were performed according to Sumiyoshi et al. with slight modifications [40]. Dry cell wall material (2 mg) was treated with endopolygalacturonase (EPG, Megazyme, Ireland), a pectin hydrolase, and the mixture was stirred for 16 h at 30°C and 40 rpm in a stirrer as described in T. Ishii et al. [41]. The sample was then centrifuged, and the supernatant was used as the EPG fraction, the pellet was hydrolyzed in 2 M trifluoroacetic acid (TFA; Wako, Osaka, Japan) at 121°C for 2 h, and the pellets were hydrolyzed in 72% H2SO4 at room temperature for 2 h. Each fraction was treated with methanol: hydrogen chloride, and the resulting methyl glycosides were converted to trimethylsilyl derivatives and analyzed by gas-liquid chromatography (GC-2010; Shimadzu, Kyoto, Japan). The hexose sugar content and uronic acid content were determined by the anthrone method and the meta-hydroxybiphenyl method, respectively.

  1. Authors have introduced two genes, STAR1 and OsXTH19, but only described some traits or performances of mutant lines from star1, not clearly analyzed the results. In addition, there was no crosslinking about two genes in results or experiments.

According to the reviewer’s advice, we have revised the Results and Discussion sections. In the discussion section especially, we added thematic paragraphs and subheadings to organize the content.

  1. The “Monosaccharide composition” were widely differencing between two experiments, see figure 3 and figure 4, the WT lines would be same or almost same, but they were some big differences between two detections, please check them.

Many thanks for your comment. The WT of Figure 3 is cv Koshihikari. The WT of Figure 4 is cv Nipponbare. We added sentences to the Methods section and the figure legends for Figures 3 and 4.

(L300)

  1. Materials and Methods

4.1. Plant Material and Growth Conditions

The wild-type (Oryza sativa cv. Koshihikari) and the star1 mutant (Koshihikari background) [15] were used in the experiments. The wild-type (Oryza sativa cv. Nipponbare) and the OsXTH19-OX (Nipponbare background, OX15 line) [17] rice strains were also used in the experiments.

(L136)

Figure 3. Monosaccharide composition of TFA-soluble fractions of WT and the aluminum-sensitive mutant star1.Monosaccharide composition of TFA-soluble fractions in roots from WT (cv Koshihikari) and star1. Plants were grown in 1.0 mM CaCl2 at pH 4.5 and then treated without or with Al (0: -Al or 100 µM AlCl3: +Al). SD is given in parentheses. Different letters in each panel indicate significant differences at p < 0.05 (Tukey's test). [Data represent means of independent biological replicates ± standard deviation. n = 6 (WT) and n = 6 (star1)].

(L166)

Figure 4. Monosaccharide composition of TFA-soluble fractions of WT and OsXTH19-OX. Monosaccharide composition of TFA-soluble fractions in roots from WT (cv Nipponbare) and OsXTH19-OX. SD is given in parentheses. Plants were grown in 1.0 mM CaCl2 at pH 4.5 and then treated without or with Al (0: -Al or 100 µM AlCl3: +Al). Different letters in each panel indicate significant differences at p < 0.05 (Tukey's test). [Data represent means of independent biological replicates ± standard deviation. n = 6 (WT) and n = 7(OsXTH19-OX)].

  1. In Pectin staining, the star1 mutant lines were missing, it would be important supplements for authors suppose.

The results of pectin staining, the star1 mutant lines have already published in Reference No 14.  Nagayama, T.; Tatsumi, A.; Nakamura, A.; Yamaji, N.; Satoh, S.; Furukawa, J.; Iwai, H. Effects of Polygalacturonase Overexpression on Pectin Distribution in the Elongation Zones of Roots under Aluminium Stress. AoB PLANTS 2022, 14, plac003, doi:10.1093/aobpla/plac003.

So, we described about the pectin in the star1 mutant in discussion as follows.

(L288)

In contrast, in star1, which has low Al tolerance, pectin was not secreted from the roots even when exposed to Al stress, and Al accumulated in the cell wall [14, 37,38].

  1. Authors thought OsXTH19-OE would reduce Al accumulation by transporting it, such as STAR1 gene, authors could detect its expression in WT and OsXTH19-OE lines to confirm its function pathway.

We have revised the Results and Discussion sections. In the discussion section especially, we added thematic paragraphs and subheadings to organize the content.

  1. “star1 mutant” in line 59 should be italic, etc.

Thank you for your advice. We have made the necessary changes to the manuscript and have changed the text to italics in red.

  1. Some description in figures could be removed to method section, such as “Plants were grown in 1.0 mM CaCl2 at pH 4.5 and then treated without or with Al (0: -Al or 100 μM AlCl3: +Al),”, etc.

 We deleted this description in figure legends.

We believe our manuscript now meets standards for publication. I am looking forward to hearing from you soon.

Sincerely yours,

Hiroaki Iwai

Tokai University

Department of Biology, School of Biological Sciences,

Sapporo, Hokkaido 005-8601, Japan

iwai.hiroaki.gb@tokai.ac.jp

Reviewer 3 Report

Comments and Suggestions for Authors

Review of the manuscript 'OsXTH19 overexpression improves aluminium tolerance via xyloglucan reduction in rice root cell wall.'
The manuscript addresses the mechanisms of tolerance to aluminium (Al) toxicity in rice (Oryza sativa), with a focus on the role of hemicellulose and xyloglucans in root cell walls. The authors analyse the effect of overexpressing the OsXTH19 gene, which encodes the xyloglucan endotransglucosylase/hydrolase (XTH) enzyme. This affects aluminium accumulation and root growth under aluminium stress. The work is based on correctly performed biochemical and physiological experiments, which are adequately documented. The topic is timely and relevant to the issue of abiotic tolerance in crops on acidic soils. While the manuscript has potential for publication, it requires significant revisions.

Introduction:
The introduction provides basic information on aluminium (Al) toxicity and its effects on plant growth, offering an adequate background to the research problem presented. The authors correctly emphasise that the primary site of aluminium accumulation in plant cells is the cell wall, particularly the outer layers of root cells, which inhibits cell elongation. However, a deeper characterization of the biochemical function of cell wall components (e.g., pectins, hemicellulose, and cellulose) and their interaction with aluminum is lacking — the current coverage is too brief. Additionally, the introduction should include a description of the role of XTH in cell wall modification in the context of abiotic stress, rather than just mechanical loosening of the wall, to justify the selection of OsXTH19 as an overexpressed gene.
Materials and Methods
There is a lack of information on the number of seeds used per trial and the number of biological replicates performed in the experiment described in Section 4.1. In Section 4.2, the exact conditions and concentration of the EPG enzyme (e.g., amount, unit activity, and manufacturer) are not specified. Information on the incubation time, temperature, and buffer used in TFA and H₂SO₄ digestion is also missing.

Discussion:
Overall, the discussion's structure is logical. However, in places, the argument is chaotic and overly elaborate, with no clear division into subtopics. It is recommended that thematic paragraphs or subheadings be used to organize the content. There was no mention of temporal differences (short-term vs. long-term effects of Al), which is essential for localising its toxic effects. Too much space was devoted to the mechanism of action of XTH. I suggest shortening this section and moving it to the introduction, rather than repeating it. There is no mention of the possibility that other components of hemicelluloses can also bind aluminium. I think this should at least be suggested.

Comments on the Quality of English Language

Numerous grammatical and syntactical errors throughout the text.

Author Response

Dear Reviewer 3,

We are grateful to the editor and reviewers for their constructive and helpful comments, which greatly helped us improve our manuscript titled “OsXTH19 Overexpression Improves Aluminum Tolerance via Xyloglucan Reduction in Rice Root Cell Wall” (plants-3689902). As indicated in the following responses, we have considered all these comments in preparing this resubmitted manuscript. Detailed responses to each comment are listed below.

Reviewer 3

The manuscript addresses the mechanisms of tolerance to aluminium (Al) toxicity in rice (Oryza sativa), with a focus on the role of hemicellulose and xyloglucans in root cell walls. The authors analyse the effect of overexpressing the OsXTH19 gene, which encodes the xyloglucan endotransglucosylase/hydrolase (XTH) enzyme. This affects aluminium accumulation and root growth under aluminium stress. The work is based on correctly performed biochemical and physiological experiments, which are adequately documented. The topic is timely and relevant to the issue of abiotic tolerance in crops on acidic soils. While the manuscript has potential for publication, it requires significant revisions.

Thank you for your review. We have thoroughly revised our manuscript, focusing on the introduction and discussion points you raised, and had a native English speaker proofread the manuscript to ensure clarity for "Plants" readers.

Introduction:
The introduction provides basic information on aluminium (Al) toxicity and its effects on plant growth, offering an adequate background to the research problem presented. The authors correctly emphasise that the primary site of aluminium accumulation in plant cells is the cell wall, particularly the outer layers of root cells, which inhibits cell elongation. However, a deeper characterization of the biochemical function of cell wall components (e.g., pectins, hemicellulose, and cellulose) and their interaction with aluminum is lacking — the current coverage is too brief. Additionally, the introduction should include a description of the role of XTH in cell wall modification in the context of abiotic stress, rather than just mechanical loosening of the wall, to justify the selection of OsXTH19 as an overexpressed gene.

Thank you for your comments. We fully agree with your advice. According to the reviewer’s advice, we have revised the Introduction sections as follows.

(L45)

Previous studies have reported that the cell wall is the major site of Al accumulation [9]. The cell wall is the first site of contact with Al and plays an important role not only in the control of plant development, but also in the recognition and tolerance of Al toxicity. For example, Clarkson reported that 85% to 90% of the total Al accumulated in barley roots is tightly bound to cell walls[10]. The main components of plant cell walls are three polysaccharides: cellulose, hemicellulose, and pectin. These polysaccharides determine cell shape and mechanical strength. The cell wall matrix, which includes pectin and hemicellulose polysaccharides, generally occupies 10–30% of the dry weight of plants. Pectin acts as a cement, binding plant cells together, while hemicellulose forms cross-links between cellulose fibers. Pectin was thought to be the major polysaccharide capable of binding Al because it is negatively charged and contains carboxyl groups that can directly bind Al [11]. However, in recent years, several studies have reported that it is hemicellulose that contributes significantly to the Al-binding capacity of cell walls [12]. This study suggests that O-acetylation of xyloglucan in hemicellulose affects its Al-binding capacity, thereby influencing Al sensitivity [12]. Therefore, the results imply that one key mechanism by which plants cope with Al stress is by decreasing the Al content of cell walls through altering their components and properties. However, in rice, it is not clear where and how much Al accumulates in the cell wall and how this leads to growth inhibition. The mechanisms by which root-absorbed Al binds to cell wall components and what changes in cell wall components cause the inhibition of root elongation are still unclear. In this study, we used rice (Oryza sativa), which has relatively high Al tolerance [13,14]. We analyzed the Al responsibility of WT with Al tolerance, star1 mutant as Al-sensitive rice, and xyloglucan endotransglucosylase/hydrolase (XTH) OsXTH19-OXas rice with suppressed xyloglucan accumulation. The STAR1 gene encodes an ABC transporter, and its expression is controlled by the ART1 transcription factor[15], which increases expression in response to Al stress. We analyzed the Al liability of WT (cv Koshihikari) with Al tolerance, star1 mutant as Al sensitive rice, and one of the xyloglucan endotransglucosylase/hydrolase (XTH) family OsXTH19-OX as rice with suppressed xyloglucan accumulation. Since the star1 mutant is strongly inhibited by Al, these genes are thought to contribute to Al tolerance in rice [16]. OsXTH19 has only xyloglucan endohydrolase (XEH) activity and no endotransglucosylase (XET) activity [17]. Therefore, in OsXTH19-OX, xyloglucan degradation proceeds, suppressing its accumulation and content. The XTH family is divided into three groups (I, II, and III) based on the structural features of each individual [14]. Group III comprises two distinct clades: Group III-A and Group III-B. According to crystallography data, Group III-A members are predicted to exhibit hydroase activity rather than XET activity. OsXTH19, a Group III-A member, is specifically expressed in the elongating zones of roots [17]. Thus, this study focuses on the effect of aluminum on root elongation and its relationship to OsXTH19. In this study, we examined changes in Al accumulation sites and cell wall components in rice to elucidate the mechanism by which Al inhibits growth.

Materials and Methods
There is a lack of information on the number of seeds used per trial and the number of biological replicates performed in the experiment described in Section 4.1. In Section 4.2, the exact conditions and concentration of the EPG enzyme (e.g., amount, unit activity, and manufacturer) are not specified. Information on the incubation time, temperature, and buffer used in TFA and H₂SO₄digestion is also missing.

Thank you for your comments. According to the reviewer’s advice, we have revised the Methods sections as follows.

  1. Materials and Methods

4.1. Plant Material and Growth Conditions

The wild-type (Oryza sativa cv. Koshihikari) and the star1 mutant (Koshihikari background) [15] were used in the experiments. The wild-type (Oryza sativa cv. Nipponbare) and the OsXTH19-OX (Nipponbare background, OX15 line) [17] rice strains were also used in the experiments. Seedlings were submerged in ion exchange water at 30 °C for 3 days, followed by 3 days in 1.0 mM CaCl2, pH 4.5. Grown seedlings were treated with or without Al for 1 day. Free Al activity was determined by GEOCHEM-EZ software [39], ranging between 76.57% to 78.28%. Plants were grown at 30 °C under continuous light of 250 μmol m-2 s-1. To calculate root elongation during Al treatment, root length was measured before and after Al treatment using a ruler. The relative root elongation, RRE (%) = (root growth in each Al condition)/(root growth in the control) × 100, was calculated to compare the root elongation and the Al tolerance between the different plants. All experiments were performed at least five times with independent biological replicates.

4.2. Cell Wall Analysis

Root tips (0-1 mm) from three seedlings were cut with a razor and collected in a 2.0 ml tube as a cell wall sample. Samples were frozen in liquid nitrogen and crushed with a pestle. Cell wall extraction and analysis were performed according to Sumiyoshi et al. with slight modifications [40]. Dry cell wall material (2 mg) was treated with endopolygalacturonase (EPG, Megazyme, Ireland), a pectin hydrolase, and the mixture was stirred for 16 h at 30°C and 40 rpm in a stirrer as described in T. Ishii et al. [41]. The sample was then centrifuged, and the supernatant was used as the EPG fraction, the pellet was hydrolyzed in 2 M trifluoroacetic acid (TFA; Wako, Osaka, Japan) at 121°C for 2 h, and the pellets were hydrolyzed in 72% H2SO4 at room temperature for 2 h. Each fraction was treated with methanol: hydrogen chloride, and the resulting methyl glycosides were converted to trimethylsilyl derivatives and analyzed by gas-liquid chromatography (GC-2010; Shimadzu, Kyoto, Japan). The hexose sugar content and uronic acid content were determined by the anthrone method and the meta-hydroxybiphenyl method, respectively.

Discussion:
Overall, the discussion's structure is logical. However, in places, the argument is chaotic and overly elaborate, with no clear division into subtopics. It is recommended that thematic paragraphs or subheadings be used to organize the content. There was no mention of temporal differences (short-term vs. long-term effects of Al), which is essential for localising its toxic effects. Too much space was devoted to the mechanism of action of XTH. I suggest shortening this section and moving it to the introduction, rather than repeating it. There is no mention of the possibility that other components of hemicelluloses can also bind aluminium. I think this should at least be suggested.

Following the reviewer’s advice, we revised the discussion sections. We added thematic paragraphs and subheadings to organize the content. We also deleted some redundant descriptions.

  1. Discussion

3.1. Reevaluating the Primary Target of Aluminum Toxicity and the Role of Hemicellulose in Rice Cell Walls

Whether the main target of Al toxicity is the cell wall or intracellular is still a matter of debate [5,18]. Since Al has a rather strong binding affinity for enzymatic reactions [5], Al likely targets multiple sites simultaneously. On the other hand, given that the cell wall is the main site of Al accumulation, it seems reasonable that the cell wall is the main target site of Al toxicity, especially when the treatment period is short. Indeed, there is ample experimental evidence that Al-induced root growth inhibition is associated with disruption of cell wall function [18]. For example, Al treatment has been shown to affect the growth of many plant species such as pumpkin [9], wheat [19], and rice [12–14]. Li et al. reported that the mechanism of Al-induced root growth inhibition in maize is the binding of Al to pectin [7]. However, the biochemical analysis of the cell wall underlying the inhibition of root elongation in response to Al treatment is not clear. Therefore, pectin has long been recognized as the major component of Al binding that inhibits root elongation, but as shown in the study of Nagayama et al. Our results indicate that hemicellulose is the major contributor to Al accumulation in the rice cell wall.

3.2. Accumulation of Aluminum and Calcium in the Cell Wall and Their Roles in Aluminum Tolerance

Low Al tolerance strains and the star1 mutant have been reported to accumulate more Al [15,20], and our results were similar (Figure 1, 7). The background level of Al measurement derived from seeds that plants have before Al treatment is considered to be in the range of 0 to 1 µg per 1 mg of dry cell wall. About half of the Al present in the cell wall was contained in the TFA-soluble fraction, which is rich in hemicellulose (Figure 1). Several previous studies have also reported that hemicellulose adsorbs Al [14,23], suggesting that Al absorbed by plants is mainly accumulated in the hemicellulose region of the cell wall. Some of the hydrogen atoms contained in the polysaccharides of the hemicellulose form hydrogen bonds with cellulose, thereby cross-linking the cellulose in the hemicellulose. Al3+ can accumulate by forming hydrogen bonds with hydroxyl groups in cellulose and hemicellulose regions without side chains. Even in Arabidopsis [12] and pumpkin [9], which have lower hemicellulose content than rice, hemicellulose has been confirmed as the major Al accumulation site, and its role in Al tolerance can be considered important. The EPG fraction, which contains a large amount of pectin, accounted for 30 to 40% of the Al accumulated in the cell wall (Figure 1). The main chain of pectin consists mainly of galacturonic acid, a type of uronic acid. It is then demethylated by pectin methylesterase (PME), exposing the carboxyl group of the galacturonic acid, which is thought to become negatively charged, making it easier to bind to cations such as Ca2+ [20]. Normally, carboxyl groups gel by forming a cross-linked structure with Ca2+, which imparts viscoelasticity to the cell wall [22]. However, pectin has a higher affinity for Al3+ than for Ca2+, and Al3+ also has a high adsorption capacity for pectin [23]. Ca is an essential macronutrient for plants and plays an important role in plant development and stress responses as a structural component in cell walls and membranes, and as an intracellular messenger in the cytoplasm [24]. It has been shown that Ca functions to reduce oxidative stress and alleviate heavy metal toxicity [4,10,25], and that elongation of rice roots and aerial parts is significantly suppressed under non-Ca-supplemented conditions [26]. Previous studies have observed that Al treatment inhibits Ca uptake and significantly reduces the amount of Ca transported to aerial parts [26]. These results suggest that under Al stress conditions, Ca transport to the aerial parts was inhibited so that both Al and Ca were present in large amounts. In star1, where more Ca was detected in roots than in WT, it is possible that Ca transport was more inhibited than in WT (Figure 2).

3.3. Xyloglucan Accumulation and Its Role in Aluminum Binding and Cell Wall Modification in Rice

Twenty to thirty percent of the Al accumulated in the cell walls was contained in the TFA-insoluble fraction, which contains a large amount of cellulose (Figure 1). Cellulose is a linear polysaccharide highly condensed with β-D-glucose. This main chain forms microfibrils through intramolecular and intermolecular hydrogen bonds, giving it a strong structure. As in the case of hemicellulose, Al3+ is thought to accumulate by forming hydrogen bonds with cellulose. However, because the bonds between celluloses are strong, there are few binding sites for Al3+ to cellulose, which may have reduced the amount of Al accumulated in cellulose. Hemicellulose in grasses is mainly composed of xylan, Mix-linked-β-D-glucan (M LG), and xyloglucan. Hemicellulose in grasses is mainly composed of xylan, MLG, and xyloglucan, each with distinct sugar compositions. [27, 28]. Analysis of the constituent sugars of the TFA-soluble fraction containing a large amount of hemicellulose showed that xylose, glucose, and glucuronic acid were the major components (Figure 3, 4). It also contained a large amount of galacturonic acid, which is the main component of pectin, and it is suggested that the TFA-soluble fraction contained pectin with high binding properties to cell wall (Figure 3). Among the constituent sugars of hemicellulose in pumpkin roots, an increase in the amount of glucose and xylose was observed due to Al treatment, which is thought to contribute to the increase in xyloglucan content [9,29]. These results suggest that Al stress increases xyloglucan in hemicellulose. XTH is an important enzyme involved in xyloglucan metabolism (Figure 4). The XTH enzyme is involved in cell wall elongation through endohydrolase (XEH) activity, which hydrolyzes xyloglucan, and endotransglucosylase (XET) activity, which rearranges xyloglucan [17,30]. In Arabidopsis mutants in which XTH31, which contributes significantly to xyloglucan transfer activity, is deleted and xyloglucan accumulation is suppressed, the amount of Al accumulated in roots is reduced, and high Al tolerance is obtained. [21]. It has also been reported that when XET activity is inhibited by Al stress, the cleavage of the xyloglucan backbone and the subsequent binding of new xyloglucan chains are prevented, and cell wall elongation is inhibited [12]. While xyloglucan is the major component in the cell wall of dicots, the amount of xyloglucan in the rice cell wall is very low. It has been discussed whether xyloglucan is, in actuality, the substrate for all XTH in monocots [17,31–33]. Although xyloglucan and XTH enzymes are important for Al toxicity in dicotyledons, which contain a large amount of xyloglucan, little is known about them in rice. Despite the difference in the amount of xyloglucan between monocots and dicots, rice is known to have the same amount of XTH genes as Arabidopsis [34], and xyloglucans may also play an important role in regulating cell wall properties in rice. Xyloglucan is thought to act as a tether between cellulose microfibrils in the primary cell wall, limiting cell wall loosening [33,34]. However, although endoglucanase, a xyloglucan-specific degrading enzyme, degraded most of the xyloglucan in the cell wall, the cell wall did not loosen [17]. Only enzymes capable of degrading both xyloglucan and cellulose were effective in loosening the cell wall [32,33]. These results suggest that xyloglucans are intertwined with cellulose microfibrils, forming regions that require enzymes with both cellulase and xyloglucanase activities to degrade. In addition, small amounts of xyloglucan adhere to adjacent cellulose microfibrils, suggesting that cell wall extensibility may be controlled in this limited region (biomechanical hotspot)[35]. Furthermore, observations of NMR spectra have confirmed that a complex is formed between xyloglucan and Al, and that Al can bind to xyloglucan [36]. OsXTH19 used in this study has only hydrolytic activity. Therefore, in OsXTH19-OX, the degradation of xyloglucan proceeds, suppressing its accumulation and content (Figure 4). In this OsXTH19-OX, the elongation inhibition by Al was suppressed (Figure 6), and the amount of Al accumulated tended to be low, suggesting that xyloglucan is the target site of Al (Figure 7).

3.4. Reduced Xyloglucan Content Enhances Aluminum Tolerance Despite Lower Pectin Levels

Nagayama et al. reported that rice varieties with high Al tolerance increase the amount of water-soluble pectin from their roots when they receive Al. This pectin acts as a barrier and reduces toxicity by preventing Al from adsorbing to hemicellulose, which is the active site of Al and one of the cell wall components. In contrast, in star1, which has low Al tolerance, pectin was not secreted from the roots even when exposed to Al stress, and Al accumulated in the cell wall [14, 37,38]. However, in OxXTH19-OX, which had a higher tolerance for Al than the WT, the amount of pectin was less than that of the WT when treated with Al (Figure 4, 5). It was suggested that even if there was not a lot of pectin, Al tolerance was high when the amount of xyloglucan, the target of Al, was low. Since the build-up of Al in the roots was blocked in OsXTH19-OX, which has a low amount of xyloglucan, it is possible that the direct build-up of Al in xyloglucan affects cell wall growth and stops root growth. Factors important for cell growth that work by binding to xyloglucan, such as XTH and expansin, may not work as well when Al is present in xyloglucan. It has also been reported that monocotyledonous plants can tolerate Al [13,38]. This suggests that one reason for this is that the cell walls of monocotyledonous plants have a lower amount of xyloglucan compared to the cell walls of dicotyledonous plants.

We believe our manuscript now meets standards for publication. I am looking forward to hearing from you soon.

Sincerely yours,

Hiroaki Iwai

Tokai University

Department of Biology, School of Biological Sciences,

Sapporo, Hokkaido 005-8601, Japan

iwai.hiroaki.gb@tokai.ac.jp

Round 2

Reviewer 1 Report

Comments and Suggestions for Authors

The authors have made satisfactory corrections to most of my previous comments. I have a few minor clarifications and changes which the authors should incorporate for the final version.

  1. The authors explain in abstract and methods that OsXTH19-OX indicates overexpression of the OsXTH19 gene and this could also be added in the Introduction at first mention of OsXTH19-OX.
  2. For the sentence "About 2.3 times more Ca was detected in star1 than in the control under Al stress conditions." specify what the "control" denotes. Is this a control treatment or control germplasm (eg WT)? Also make sure the difference is statistically significant. 
  3. Figure 6 legend states:

    "Root elongation differed significantly between WT and OsXTH19-OX rice under 50 and 100 µM Al treatment."  However, at 50 uM A both bars have a "b" indicating no significant difference- please correct. 

  4. Take care to use acronyms once defined. The authors continue to spell out 

    "inductively coupled plasma-atomic emission spectrometry" (legends of Figures 1 and 2) despite having defined it earlier in the main text as ICP-AES.

Author Response

Dear Reviewer 1,

We are grateful to the editor and reviewers for their constructive and helpful comments, which greatly helped us improve our manuscript titled “OsXTH19 Overexpression Improves Aluminum Tolerance via Xyloglucan Reduction in Rice Root Cell Wall” (plants-3689902). As indicated in the following responses, we have considered all these comments in preparing this resubmitted manuscript. Detailed responses to each comment are listed below.

Reviewer 1

The authors have made satisfactory corrections to most of my previous comments. I have a few minor clarifications and changes which the authors should incorporate for the final version.

 Thank you for your review. We fully agree Reviewer 1’s comments.

  1. The authors explain in abstract and methods that OsXTH19-OX indicates overexpression of the OsXTH19 gene and this could also be added in the Introduction at first mention of OsXTH19-OX.

According to the reviewer’s comments, we added the sentences in Introduction as follows.

(L72)

, and one of the xyloglucan endotransglucosylase/hydrolase (XTH) family OsXTH19-OX, which overexpresses OsXTH19, as rice with suppressed xyloglucan accumulation.

  1. For the sentence "About 2.3 times more Ca was detected in star1 than in the control under Al stress conditions." specify what the "control" denotes. Is this a control treatment or control germplasm (eg WT)? Also make sure the difference is statistically significant. 

Thank you for your advice. We have corrected the errors as follows.

(L109)

There was a tendency for more Ca was detected in star1 than in the WT under Al stress conditions.

  1. Figure 6 legend states:

"Root elongation differed significantly between WT and OsXTH19-OX rice under 50 and 100 µM Al treatment."  However, at 50 uM A both bars have a "b" indicating no significant difference- please correct. 

Thank you for your advice. We have corrected the errors as follows.

(L180)

Root elongation differed significantly between WT and OsXTH19-OX rice under 100 µM Al treatment (P < 0.05, Student’s t-test). Data are means ± SDs, n = 12.

  1. Take care to use acronyms once defined. The authors continue to spell out 

"inductively coupled plasma-atomic emission spectrometry" (legends of Figures 1 and 2) despite having defined it earlier in the main text as ICP-AES.

Thank you for your advice. We have corrected the errors as follows.

Figure 1. Al content in different cell wall fractions of WT and the aluminum-sensitive mutant star1. Plants were treated without or with Al (0: -Al or 100 µM AlCl3: +Al), and cell wall fractions were extracted from roots and fractionated into different residues (endo-polygalacturonase (EPG)-soluble fraction as pectic fraction, trifluoroacetic acid (TFA)-soluble fraction as hemicellulosic fraction, and TFA-insoluble fraction as cellulosic fraction). After extraction of Al with HNO3, Al content was determined by ICP-AES. Data represent means of independent biological replicates ± standard deviation (n = 5). Different letters in each panel indicate significant differences at p < 0.05 (Tukey's test). ND, not detected.

Figure 2. Ca content in different cell wall fractions of WT and the aluminum-sensitive mutant star1. Plants were treated without or with Al (0: -Al or 100 µM AlCl3: +Al), and cell wall fractions were extracted from roots and fractionated into different residues (EPG-soluble fraction as pectic fraction and TFA-soluble fraction as hemicellulosic fraction). After the extraction of Al with HNO3, Al content was determined by ICP-AES. Data represent means of independent biological replicates ± standard deviation (n = 5). Different letters in each panel indicate significant differences at p < 0.05 (Tukey's test).

We believe our manuscript now meets standards for publication. I am looking forward to hearing from you soon.

Sincerely yours,

Hiroaki Iwai

Tokai University

Department of Biology, School of Biological Sciences,

Sapporo, Hokkaido 005-8601, Japan

iwai.hiroaki.gb@tokai.ac.jp

Reviewer 2 Report

Comments and Suggestions for Authors

Thanks for authors work, my comments were well addressed in revision. But the figure 5 was not well displayed in PDF file, please check it. Good luck.

Author Response

Dear Reviewer 2,

We are grateful to the editor and reviewers for their constructive and helpful comments, which greatly helped us improve our manuscript titled “OsXTH19 Overexpression Improves Aluminum Tolerance via Xyloglucan Reduction in Rice Root Cell Wall” (plants-3689902). As indicated in the following responses, we have considered all these comments in preparing this resubmitted manuscript. Detailed responses to each comment are listed below.

Reviewer 2

Thanks for authors work, my comments were well addressed in revision. But the figure 5 was not well displayed in PDF file, please check it. Good luck.

 Thank you for your review. We revised Figure 5 and confirmed that it is displayed correctly.

We believe our manuscript now meets standards for publication. I am looking forward to hearing from you soon.

Sincerely yours,

Hiroaki Iwai

Tokai University

Department of Biology, School of Biological Sciences,

Sapporo, Hokkaido 005-8601, Japan

iwai.hiroaki.gb@tokai.ac.jp

Reviewer 3 Report

Comments and Suggestions for Authors

I am satisfied with the changes made to the manuscript. However, the number of seeds/seedlings used in each combination is missing. In discussion, I suggest further shortening the section on XTH catalysis. Again, I recommend discussing the effects of long-term Al toxicity (e.g., beyond 48 hours) or clarifying the time limitations of the study.

Author Response

Dear Reviewer 3,

We are grateful to the editor and reviewers for their constructive and helpful comments, which greatly helped us improve our manuscript titled “OsXTH19 Overexpression Improves Aluminum Tolerance via Xyloglucan Reduction in Rice Root Cell Wall” (plants-3689902). As indicated in the following responses, we have considered all these comments in preparing this resubmitted manuscript. Detailed responses to each comment are listed below.

Reviewer 3

I am satisfied with the changes made to the manuscript.

Thank you for your review.

However, the number of seeds/seedlings used in each combination is missing.

We have described about the number as follows.

Figure 1. Al content in different cell wall fractions of WT and the aluminum-sensitive mutant star1. Plants were treated without or with Al (0: -Al or 100 µM AlCl3: +Al), and cell wall fractions were extracted from roots and fractionated into different residues (endo-polygalacturonase (EPG)-soluble fraction as pectic fraction, trifluoroacetic acid (TFA)-soluble fraction as hemicellulosic fraction, and TFA-insoluble fraction as cellulosic fraction). After extraction of Al with HNO3, Al content was determined by ICP-AES. Data represent means of independent biological replicates ± standard deviation (n = 5). Different letters in each panel indicate significant differences at p < 0.05 (Tukey's test). ND, not detected.

Figure 2. Ca content in different cell wall fractions of WT and the aluminum-sensitive mutant star1. Plants were treated without or with Al (0: -Al or 100 µM AlCl3: +Al), and cell wall fractions were extracted from roots and fractionated into different residues (EPG-soluble fraction as pectic fraction and TFA-soluble fraction as hemicellulosic fraction). After the extraction of Al with HNO3, Al content was determined by ICP-AES. Data represent means of independent biological replicates ± standard deviation (n = 5). Different letters in each panel indicate significant differences at p < 0.05 (Tukey's test).

Figure 3. Monosaccharide composition of TFA-soluble fractions of WT and the aluminum-sensitive mutant star1.Monosaccharide composition of TFA-soluble fractions in roots from WT (cv Koshihikari) and star1. Plants were treated without or with Al (0: -Al or 100 µM AlCl3: +Al). SD is given in parentheses. Different letters in each panel indicate significant differences at p < 0.05 (Tukey's test). [Data represent means of independent biological replicates ± standard deviation. n = 6 (WT) and n = 6 (star1)].

Figure 4. Monosaccharide composition of TFA-soluble fractions of WT and OsXTH19-OX. Monosaccharide composition of TFA-soluble fractions in roots from WT (cv Nipponbare) and OsXTH19-OX. SD is given in parentheses. Plants were treated without or with Al (0: -Al or 100 µM AlCl3: +Al). Different letters in each panel indicate significant differences at p < 0.05 (Tukey's test). [Data represent means of independent biological replicates ± standard deviation. n = 6 (WT) and n = 7(OsXTH19-OX)].

Figure 5. Pectin staining in the roots of WT and OsXTH19-OX seedlings treated without or with Al (0 or 100 µM AlCl3). Roots were stained with 0.01 % ruthenium red for 5 min after saponification (0.1 N NaOH, 1 min). The experiments were performed at least five times with similar results. Bars = 0.1 mm.

Figure 6. The effect of Al on root growth of WT and OsXTH19-OX. Plants were treated with Al (0, 50, 100 µM AlCl3). Seedling root length was measured before and after Al treatment, and the amount of root elongation was determined. Root elongation differed significantly between WT and OsXTH19-OX rice under 100 µM Al treatment (P < 0.05, Student’s t-test). Data are means ± SDs, n = 12.

Figure 7. Al content in different cell wall fractions of WT and OsXHT19-OX. Plants were treated without or with Al (0: -Al or 100 µM AlCl3: +Al), and cell wall fractions were extracted from roots and fractionated into different residues (endo-polygalacturonase (EPG)-soluble fraction as pectic fraction, trifluoroacetic acid (TFA)-soluble fraction as hemicellulosic fraction, and TFA-insoluble fraction as cellulosic fraction). After extraction of Al with HNO3, Al content was determined by inductively coupled plasma-atomic emission spectrometry. Data represent means of independent biological replicates ± standard deviation (n = 5). Different letters in each panel indicate significant differences at p < 0.05 (Tukey's test).

In discussion, I suggest further shortening the section on XTH catalysis.

We shortened paragraph 3.3 of the Discussion section by removing redundant and overlapping sentences, reducing its length from 645 words to 589 as follows.

                                          (L241)

3.3. Xyloglucan Accumulation and Its Role in Aluminum Binding and Cell Wall Modification in Rice

Twenty to thirty percent of the Al accumulated in the cell walls was contained in the TFA-insoluble fraction, which contains a large amount of cellulose (Figure 1). Cellulose is a linear polysaccharide highly condensed with β-D-glucose. This main chain forms microfibrils through intramolecular and intermolecular hydrogen bonds, giving it a strong structure. As in the case of hemicellulose, Al3+ is thought to accumulate by forming hydrogen bonds with cellulose. However, because the bonds between celluloses are strong, there are few binding sites for Al3+ to cellulose, which may have reduced the amount of Al accumulated in cellulose. Hemicellulose in grasses is mainly composed of xylan, MLG, and xyloglucan, each with distinct sugar compositions. [27, 28]. Analysis of the constituent sugars of the TFA-soluble fraction containing a large amount of hemicellulose showed that xylose, glucose, and glucuronic acid were the major components (Figure 3, 4). It also contained a large amount of galacturonic acid, which is the main component of pectin, and it is suggested that the TFA-soluble fraction contained pectin with high binding properties to cell wall (Figure 3). Among the constituent sugars of hemicellulose in pumpkin roots, an increase in the amount of glucose and xylose was observed due to Al treatment, which is thought to contribute to the increase in xyloglucan content [9,29]. These results suggest that Al stress increases xyloglucan in hemicellulose. XTH is an important enzyme involved in xyloglucan metabolism (Figure 4). The XTH enzyme is involved in cell wall elongation through endohydrolase (XEH) activity, which hydrolyzes xyloglucan, and endotransglucosylase (XET) activity, which rearranges xyloglucan [17,30]. In Arabidopsis mutants in which XTH31, which contributes significantly to xyloglucan transfer activity, is deleted and xyloglucan accumulation is suppressed, the amount of Al accumulated in roots is reduced, and high Al tolerance is obtained. [21]. It has also been reported that when XET activity is inhibited by Al stress, the cleavage of the xyloglucan backbone and the subsequent binding of new xyloglucan chains are prevented, and cell wall elongation is inhibited [12]. While xyloglucan is the major component in the cell wall of dicots, the amount of xyloglucan in the rice cell wall is very low. It has been discussed whether xyloglucan is, in actuality, the substrate for all XTH in monocots [17,31–33]. Despite the difference in the amount of xyloglucan between monocots and dicots, rice is known to have the same amount of XTH genes as Arabidopsis [34], and xyloglucans may also play an important role in regulating cell wall properties in rice. Xyloglucan is thought to act as a tether between cellulose microfibrils in the primary cell wall, limiting cell wall loosening [33,34]. However, degradation of xyloglucan alone by specific endoglucanases does not loosen the wall; only enzymes targeting both xyloglucan and cellulose are effective [32,33]. These results suggest that xyloglucans are intertwined with cellulose microfibrils, forming regions that require enzymes with both cellulase and xyloglucanase activities to degrade. In addition, small amounts of xyloglucan adhere to adjacent cellulose microfibrils, suggesting that cell wall extensibility may be controlled in this limited region (biomechanical hotspot)[35]. Furthermore, observations of NMR spectra have confirmed that a complex is formed between xyloglucan and Al, and that Al can bind to xyloglucan [36]. OsXTH19 used in this study has only hydrolytic activity. Therefore, in OsXTH19-OX, the degradation of xyloglucan proceeds, suppressing its accumulation and content (Figure 4). In this OsXTH19-OX, the elongation inhibition by Al was suppressed (Figure 6), and the amount of Al accumulated tended to be low, suggesting that xyloglucan is the target site of Al (Figure 7).

Again, I recommend discussing the effects of long-term Al toxicity (e.g., beyond 48 hours) or clarifying the time limitations of the study.

We have added the sentences in Discussion as follows.

(L207)

Al toxicity is characterized by its short-term effects. One example is the inhibition of root elongation. The results of this study were observed within 48 h of aluminum treatment. If the treatment had been carried out for a longer period, a feedback mechanism may have occurred to alleviate the effects. However, this report is limited to findings within 48 h.

We believe our manuscript now meets standards for publication. I am looking forward to hearing from you soon.

Sincerely yours,

Hiroaki Iwai

Tokai University

Department of Biology, School of Biological Sciences,

Sapporo, Hokkaido 005-8601, Japan

iwai.hiroaki.gb@tokai.ac.jp
